# Comparative Histology of C Thyrocytes in Four Domestic Animal Species: Dog, Pig, Horse, and Cattle

**DOI:** 10.3390/ani12101324

**Published:** 2022-05-23

**Authors:** Justyna Sokołowska, Anna Cywińska, Martyna Puchalska

**Affiliations:** 1Department of Morphological Sciences, Institute of Veterinary Medicine, Warsaw University of Life Sciences-SGGW, Nowoursynowska 159, 02-776 Warsaw, Poland; 2Department of Basic and Preclinical Sciences, Faculty of Biological and Veterinary Sciences, Nicolaus Copernicus University in Toruń, Lwowska 1, 87-100 Toruń, Poland; anna_cywinska@umk.pl; 3Department of Food Hygiene and Public Health Protection, Institute of Veterinary Medicine, Warsaw University of Life Sciences-SGGW, Nowoursynowska 159, 02-776 Warsaw, Poland; martyna_puchalska@sggw.edu.pl

**Keywords:** cattle, C thyrocytes, comparative morphology, dogs, horses, immunohistochemistry, pigs, thyroid gland

## Abstract

**Simple Summary:**

In this study we have proved that dogs, pigs, cattle, and horses, species belonging to four distinct families, differ in regard to microscopical characteristics of their C thyrocytes. Although the total number of C thyrocyte profiles and their localization within thyroid lobes were comparable, each of the examined species displayed their unique morphological characteristics and distribution pattern. The differences described in our study, easily recognizable microscopically, can be used as a reference material for further studies focused on C thyrocytes biology in physiological and pathological conditions.

**Abstract:**

The number, morphology, and distribution of C thyrocytes within the thyroid gland vary among species; however, studies in domestic animals are limited. In this study we compared the morphology, distribution pattern, and percentage of C thyrocytes in four domestic species: dogs, pigs, horses, and cattle. Eighty thyroid glands, 20 per species, were examined. C thyrocytes were visualized immunohistochemically with anti-calcitonin rabbit polyclonal antibody alone and combined with the periodic acid Schiff method to simultaneously visualize C thyrocytes with the basement membranes of thyroid follicles. C thyrocyte morphology varied considerably between species, from oval- (dogs) and spindle-shaped (pigs) to polymorphic (cattle and horses). Bovine C thyrocytes demonstrated cytoplasmic protrusion. C thyrocytes were located intrafolliculary (all species), epifollicularly (dogs, horses, cattle), or interfolicularly (cattle). Most porcine and bovine C thyrocytes existed individually whereas canine C thyrocytes usually formed clusters. In horses, they tended to form groups of various shapes and sizes or even rims encompassing whole follicles. In all species, the number of C thyrocyte profiles increased from the periphery to the central area of the thyroid lobe. The mean total fraction of C thyrocytes in the superficial, intermediate, and central areas were as follows: 2.55%, 8.43%, and 12.48% in dogs; 3.81%, 7.66%, and 10.79% in pigs; 1.55%, 7.44%, and 8.87% in horses; and 2.62%, 10.75%, and 12.96% in cattle. No statistical differences in the total number of C thyrocyte profiles were observed among species (8.87% in dogs, 8.58% in cattle, 7.98% in pigs, and 5.83% in horses). Our results indicated that the studied species displayed their own morphological characteristics and distribution pattern of C thyrocytes; however, total numbers of C thyrocyte profiles and their localization within the thyroid lobe are comparable.

## 1. Introduction

The thyroid gland contains two types of endocrine cells: T thyrocytes (follicular thyrocytes, thyrocytes) and C thyrocytes (C cells, parafollicular cells); however, C thyrocytes constitute only a minor proportion of the total thyroid parenchyma. The cells are responsible for maintaining calcium homeostasis by managing calcitonin production, as well as the release of various regulatory peptides involved in the local regulation of T thyrocyte activity [1].

Although the number, morphology, and distribution of C thyrocytes within the thyroid gland are known to vary between species [1,2], detailed studies regarding this topic are limited. Most reports refer to humans [3] and laboratory animals [4,5,6] and only a few have examined domestic animals. Among the latter, some have been conducted on dogs [7,8,9] and pigs [10,11]. Only individual papers have dealt with horses and other animals [12,13,14,15].

Most studies conducted on canine C thyrocytes have focused on C thyrocytes complexes [8,16,17]; these are considered to be remnants of ultimobranchial bodies: structures assumed to serve as a vehicle for transport C thyrocyte precursors to the thyroid gland in mammals [18]. The porcine studies have examined the ultrastructural characteristic of C thyrocytes in physiological and some pathological conditions [11,19,20], as well as changes in C thyrocyte secretory function during fetal development of the thyroid gland [21]. Only isolated studies have described the microscopic characteristics and numbers of porcine C thyrocytes [6,10].

Some information on equine C thyrocyte morphology is given by Blähser [22], Tanimura et al. [23], and Yanai et al. [14]; however, these do not include detailed findings on the morphological characteristics and localization of C thyrocytes in this species. Moreover, Yoshikawa et al. [15], described the distribution of C thyrocytes throughout the thyroid gland and its association with age and sex in racehorses.

No studies so far appear to have examined the morphology and localization of C thyrocytes in cattle. Moreover, no studies describe the localization of C thyrocytes in relation to thyroid follicles. To date, the positioning of C thyrocytes was only estimated based on their localization relative to T thyrocytes without the visualization of the basement membranes of thyroid follicles. The comparative histological studies subjected on C thyrocytes focused mainly on rodents [5,6] and up to date there is no extensive comparative study analyzing various aspects of C thyrocyte histology in domestic animals in the available literature. Such comprehensive work conducted on healthy animals will be valuable for future studies on C thyrocytes, especially in pathological conditions including medullary thyroid carcinomas and C thyrocyte hyperplasia. In animals of interest, both conditions have been reported [24,25,26,27,28]; however, they have not been extensively studied. Thus, the aim of the study was to compare the morphology and distribution pattern of immunohistochemically detected C thyrocytes, as well as their percentage composition of the total endocrine cell population, between four domestic animal species: cattle (*Bos taurus*), dogs (*Canis familiaris*), horses (*Equus caballus*), and pigs (*Sus scrofa domesticus*).

## 2. Materials and Methods

### 2.1. Animals and Tissue Collection

Eighty thyroid glands were included in the study, twenty per species. The following groups of animals were examined: dogs (7 females, and10 males, aged from 1-day to 16 years), pigs (20 males, aged 6–7 months), horses (11 females and 9 males, aged from 5 months to 24 years), and cattle (20 females, aged from 1 to 8, 5 years).

The study complied with Directive 2010/63/EU and the Act of the Polish Parliament dated 15 January 2015 on the protection of animals used for scientific purposes (Journal of Laws 2015, item 266). None of the animals were killed for the study: all tissue specimens were collected from animal carcasses. The thyroid glands originating from cattle, horses, and pigs were collected from freshly slaughtered animals. The canine thyroid glands were obtained during necropsy from animals euthanized in veterinary clinics due to illnesses unrelated to thyroid function, such as paraparesis, aggression, car accidents, cardiorespiratory or renal failure, or tumors of the skin or abdominal organs.

Only thyroid glands without any macroscopic lesions were included. In each case, the central part of the left thyroid lobe, sectioned transversely, was collected. We chose this part of the thyroid lobe, as according to available data in this region the C thyrocytes are the most numerous [1,29,30]. Tissue specimens were fixed in 10% neutral buffered formalin for 24 h, processed by the common paraffin technique, and cut into 3 µm sections. In each case, two tissue specimens separated from each other by a distance of 100 µm were taken for analysis.

### 2.2. Immunohistochemistry

The C thyrocytes were visualized immunohistochemically using an anti-calcitonin rabbit polyclonal antibody (Agilent Dako, Glostrup, Denmark); cross-reactivity for each examined species has been confirmed by the manufacturer and in previous studies [21,25,26,27]. All immunohistochemical procedures were performed according to the manufacturer’s protocols. Antigen unmasking was performed by microwaving (two cycles: 7 and 5 min, 700 W, in citrate buffer pH 6.0). Following this, the slides were incubated with primary antibody, diluted at 1:400, for one hour at room temperature. Antigen detection was performed using The REAL EnVision Detection System, Peroxidase/DAB+, Rabbit/Mouse visualization system (Agilent Dako). The sections were counterstained with Erlich’s hematoxylin. Negative controls were also created, in which the primary antibody was substituted with TBST buffer (Agilent Dako).

Moreover, for each species, the C thyrocytes were co-visualized with the basement membranes of the thyroid follicles. Briefly, five additional slides per species representing the thyroid gland sections with the best visualized C thyrocytes were selected. After immunohistochemical detection with anti-calcitonin antibody, these specimens were subjected to further staining by the periodic acid Schiff method (own modification related to the use of these two methods in one staining procedure). Briefly, the reaction with the primary antibody was visualized using the DAB solution (Agilent Dako), following which, the slides were rinsed with tap water for 10 min and placed in distilled water for 5 min. They were then subjected to staining by the routine periodic acid Schiff method [31]. The slides were counterstained with Harris’ hematoxylin.

### 2.3. C Thyrocytes Scoring

The C thyrocytes were quantified in three separate areas of the medial region of the thyroid lobe: superficial, central, and intermediate. The cells were counted manually in 14 randomly selected visual fields per each area (154.89 × 193.61 µm) at a microscope magnification of 600×. The visual fields were located uniformly throughout each area of interest. In each visual field, the overall number of C and T thyrocyte profiles were calculated. Following this, the results from each area were summarized, and the proportion of C thyrocyte profiles to profiles of total endocrine cell population (fraction of C thyrocytes) was calculated according to the following formula:Fraction of C thyrocytes [%] = number of C thyrocyte profiles/number of all endocrine cells’ profiles (C and T thyrocytes) × 100%

In each case, the fraction of C thyrocytes was scored in both tissue sections originating from each lobe and the mean value was calculated. The numbers were recorded for each area of the thyroid lobe and for the whole tissue section (relative total fraction of C thyrocytes).

### 2.4. Statistical Analysis

The obtained data were analyzed using Statistica 13.3 for Windows (StatSoft, Kraków, Poland). All were presented as mean values ± SD, differences in C thyrocyte distribution patterns were calculated using the Mann–Whitney U-test. Fractions of C thyrocytes were compared between the three areas of the thyroid lobe and between species. *p* ≤ 0.05 was considered significant.

## 3. Results

### 3.1. C Thyrocyte Morphology

C thyrocyte morphology varied considerably between species. In all examined species, the cellular outlines of C thyrocytes were clearly visible in individual cells but usually faded within cell clusters, especially in horses.

In dogs, the C thyrocytes were oval or polygonal in shape, with centrally-located oval or slightly irregular nuclei (Figure 1c–e).

In pigs, the C thyrocytes had a spindle shape; these were strongly elongated, with an oval nucleus located eccentrically in most cases (Figure 2g,h).

The equine C thyrocytes were polymorphic, ranging in shape from oval or polygonal to elongated or irregular. The nuclei were round or oval, depending on cell shape, and located centrally (Figure 3h–j). In some of the C thyrocytes, especially those with a spindle shape, the cytoplasm formed slender endings resembling cytoplasmic processes (Figure 3k,l); however, no separate protrusions were visible among T thyrocytes in cross-section, i.e., without any connection with the cell bodies.

In cattle, the C thyrocytes were also polymorphic, being polygonal, oval, or irregular in shape. Some possessed clearly-visible long cytoplasmic processes (Figure 4g–i); these were most apparent in cells lying individually, particularly in thyroid follicles viewed in the tangential section. Moreover, separate processes in cross-section (i.e., without any connection with the cell bodies) were also visible between T thyrocytes (Figure 4d). The C thyrocyte nuclei were round or oval and located eccentrically.

### 3.2. C Thyrocyte Distribution within Thyroid Gland Parenchyma

In all examined species, the C thyrocytes were not uniformly distributed throughout the thyroid gland parenchyma. In the narrow subcapsular region of the thyroid gland, only single C thyrocytes were observed. In deeper regions, the C thyrocytes were located unevenly: areas of thyroid gland parenchyma containing C thyrocytes were surrounded by areas either completely lacking C thyrocytes or containing only individual C thyrocytes (Figure 1a,b, Figure 2a,b, Figure 3a,b and Figure 4a,b). The specimens from horses and cattle tended to include large regions with high populations of C thyrocytes, while the pig specimens tended to have more diffuse C thyrocyte patterns than those of the other species. Generally, the size of the areas occupied by C thyrocytes increased together with the total C thyrocyte number.

The location of C thyrocytes in relation to the thyroid follicles differed between species. In the cattle specimens, the C thyrocytes were most likely in contact with the colloid, as some cells were observed to reach the apical surface of the T thyrocytes (Figure 4c). No such situation was observed in the other three species.

In dogs, the C thyrocytes mostly formed groups of 2–10 cells; however, some larger groups of approximately 10–20 cells were observed, with some exceeding 20 cells. Individual C thyrocytes were rare (Figure 1 and Figure 5). The C thyrocytes themselves were located either within or between the thyroid follicles (Figure 5); most of them were lying intrafollicularly. A smaller number of cells were found to be located outside follicles adhering to the basement membrane (i.e., an epifollicular localization); most of these were present as large groups consisting of more than 20 C thyrocytes (Figure 5a,b). In most cases, a single C thyrocyte cluster was present within the wall of a follicle (Figure 5c–e).

In pigs, the C thyrocytes were present exclusively intrafollicularly and were observed to adhere to the follicular basement membrane (Figure 6). Most of them were present as separate cells. In follicles with more numerous C thyrocytes, the cells formed “chains”, lying in very close proximity and sometimes directly contact each other with their endings (Figure 2c,d and Figure 6d,e). Rarely, small groups consisting of two or three C thyrocytes were observed (Figure 2e). The thyroid follicles possess only single C thyrocytes within the wall; however, these cells could be more numerous and located irregularly along the follicle perimeter, especially in areas richer in C thyrocytes. Usually, the C thyrocytes in one follicle tended to share the same orientation in respect to the follicle surface, i.e., they were visible either in longitudinal or cross-section. However, some were found to be present in different orientations in a single follicle (Figure 2f and Figure 6).

In horses, the C thyrocytes tended to form “chains” (Figure 3c and Figure 7d,e) or groups of various sizes. While most C thyrocyte clusters consisted of approximately two to five cells, larger groups were also observed, especially in areas richer in C thyrocytes; some of these contained considerably more than ten cells (Figure 3d–f and Figure 7a,b,i). The cells in “chains” were also usually in contact with each other; individual C thyrocytes were less common (Figure 7e). C thyrocytes were localized either within the follicular wall or between thyroid follicles, adhering to their basement membrane (Figure 7). Relatively fewer cells demonstrated an epifollicular localization, with most being either single C thyrocytes (Figure 7f–h) or large C thyrocyte clusters (Figure 7i). One thyroid follicle usually contained at least several C thyrocytes and/or C thyrocyte groups scattered around its perimeter; however, in some cases, only single cells or isolated small clusters were observed (Figure 7e). In areas with higher numbers of C thyrocytes, the C thyrocytes occupied large areas of the follicular wall, even encompassing whole follicles in the form of solid or discontinuous rims (Figure 3g and Figure 7a,c).

In cattle, the vast majority of C thyrocytes were located individually (Figure 4d and Figure 8a,b); however, some were in contact via their processes (Figure 4g,h). In some cases, small groups consisting of two to five cells were observed (Figure 4e,f and Figure 8c,f). Most C thyrocytes were located intrafollicularly (Figure 8a–c), with only single cells lying outside the follicular wall; among the latter, most were present epifollicularly (Figure 8d–f), with some present interfollicularly, i.e., without any contact with the follicular basement membrane (Figure 8g–i). The C thyrocytes lying outside the follicular wall were frequently in contact with blood capillaries (Figure 8d–i). In most cases, the C thyrocytes were scattered along the entire follicular perimeter. The number of C thyrocytes present within the follicular wall increased with the total number of C thyrocytes. However, even in areas rich in C thyrocytes, the cells remained separate, despite being more numerous and closely located. Follicles containing single C thyrocytes were less common.

### 3.3. Fractions of C Thyrocytes

For almost all cases taken from the four species included in the study, the highest fractions of C thyrocytes were observed in the central area of the medial part of the thyroid lobe, as viewed in the cross-section, followed by the intermediate and superficial areas. 

In the dog specimens, the mean fraction of C thyrocytes was 2.55 ± 5.32 in the superficial area, followed by 8.43 ± 9.29 in the intermediate area, and 12.48 ± 9.42 in the central area (Table 1). All these differences were significant (*p* ≤ 0.001). However, taking into account individual cases, the percentage differences in C thyrocyte profiles levels between the intermediate and central areas were less than 1% in three of 20 cases, and less than 2% in another two of 20 cases. In two of 20 cases, no significant differences were found between the superficial, central, and intermediate areas. The fraction of C thyrocytes in three different regions of the medial part of the canine thyroid lobe and the total fraction of C thyrocytes in particular cases are shown in Appendix A.

In the pig specimens, the mean fractions of C thyrocytes were 3.81 ± 5.12 in the superficial area, 7.66 ± 5.49 in the intermediate area, and 10.79 ± 6.15 in the central area (*p* ≤ 0.001) (Table 1). The difference in the percentage of C thyrocyte profiles in the intermediate and central areas was less than 1% in two of the 20 examined cases and less than 2% in another two of the 20 cases. The fraction of C thyrocytes in three different regions of the medial part of the porcine thyroid lobe and the total fraction of C thyrocytes in particular cases are shown in Appendix A.

In the horse specimens, the mean fraction of C thyrocytes was 1.55 ± 3.73 in the superficial area, 7.44 ± 8.00 in the intermediate area, and 8.87 ± 7.78 in the central area (*p* ≤ 0.001) (Table 1). In this species, five of 20 cases demonstrated a higher percentage of C thyrocyte profiles in the intermediate area compared to the central area; the difference in the fraction of C thyrocytes was less than 1% in two of them, less than 2% in two, and greater than 2% in the remaining one. Among the remaining 15 of 20 cases, the difference in the fraction of C thyrocytes between the central and intermediate part was less than 1% in three animals, and less than 2% in another four. The fraction of C thyrocytes in three different regions of the medial part of the equine thyroid lobe and the total fraction of C thyrocytes in particular cases are shown in Appendix A.

In the cattle specimens, the mean fraction of C thyrocytes was 2.62 ± 4.70, 10.75 ± 7.43, and 12.96 ± 7.97 for the superficial, intermediate, and central areas, respectively (Table 1). These differences were significant (*p* ≤ 0.001). In three of the 20 cases, the fraction of C thyrocytes was higher in the intermediate area than the central area; this difference was less than 1% in the two cases, and greater than 2% in the other case. Among the remaining 17 of 20 cases, the difference in C thyrocyte profiles percentage between the central and intermediate areas was less than 2% in six animals. The fraction of C thyrocytes in three different regions of the medial part of the bovine thyroid lobe and the total fraction of C thyrocytes in particular cases are shown in Appendix A.

Comparable mean total fractions of C thyrocytes were observed for the whole examined tissue section, in dogs (mean 8.87 ± 3.72), cattle (mean 8.58 ± 2.08), and pigs (mean 7.98 ± 2.94). A slightly different value was obtained in horses, i.e., 5.83 ± 2.56, however, this difference was not significant (Table 1).

## 4. Discussion

This study gives the first detailed comparison of the morphological characteristics of C thyrocytes in four species belonging to the Canidae, Suidae, Bovidae, and Equidae. All species were found to demonstrate unique C thyrocyte morphology and distribution patterns that can be easily recognized microscopically.

### 4.1. Morphological Characteristic and Distribution Pattern of C Thyrocytes

Our observations of canine C thyrocytes confirm those from previous studies, particularly that C thyrocytes are usually clustered together in this species [2,7,9]. However, some discrepancies exist regarding the number of cells observed in these groups: while most studies indicate that these clusters are very large [7,8,16], those observed in our present study usually consisted of several C thyrocytes, rarely exceeding 20 cells. This discrepancy could be explained by the fact that the previous studies [8,16] focused on remnants of ultimobranchial bodies: these have a distinct morphology compared to normal thyroid gland parenchyma and tend to have higher numbers of C thyrocytes within them and in their vicinity [32,33]. Although these structures tend to be primarily apparent in young animals [16], large clusters of C thyrocytes, representing remnants of ultimobranchial bodies, can be also observed in adult individuals [8,17]. However, in the present study, no such potential ultimobranchial remnants were found in adult (usually aged) dogs nor in the one-day puppy included in the study. In addition, no examples of follicles lined solely by C thyrocytes were identified in any of examined canine thyroid glands, unlike Kameda [7].

Some discrepancies exist regarding the morphology of C thyrocytes in pigs. In most cases, they are described as elongated [2,6,11], while some studies have reported them as being round or oval [10,19,22]. In the present study, they were found to be spindle-shaped; however, they were indeed visible as round or oval in cross-section. This discrepancy in porcine C thyrocyte morphology was also confirmed in an electron microscopy study by Young et al. [11]. Moreover, some studies have indicated that porcine C thyrocytes possess long protrusions [6,10,22]; however, this was not observed in the present study.

Our results clarified the discrepancy regarding the distribution of porcine C thyrocytes. Some authors [22] observed them lying almost exclusively separately, whereas others [6] described large accumulations of porcine C thyrocytes surrounding thyroid follicles presented along with individual C thyrocytes. Our observations indicate that porcine C thyrocytes tend to be individual, but in rare cases, they can form small groups of two or three cells. These findings are in agreement with other studies [10,11,19].

Only limited data exist regarding the morphology of equine C thyrocytes. The most informative paper, in our opinion, was a methodological study by Blähser [22] focusing on the visualization of C thyrocytes in various mammals. The paper, however, contains a very little description of equine C thyrocytes. Further information is given by Tanimura et al. [23] on C thyrocytes within the parathyroid gland, and in figures presented by Yoshikawa et al. [15]. Our results indicate that equine C thyrocytes are polymorphic. A detailed examination of the figures given by Blähser [22] and Yoshikawa et al. [15] confirmed that in fact, they were in line with our findings. However, Blähser [22] described C thyrocytes as star-shaped, whereas a considerable proportion of the C thyrocytes observed in the present study were found to be elongated.

Blähser [22] and Tanimura et al. [23] reported that equine C thyrocytes contained cytoplasmic protrusions. Our present observations found many of the equine C thyrocytes to be irregular in shape, and although this resulted in some elongations in their cytoplasm, most did not appear to be true processes. They were not so evident, shorter, and less frequent than that observed in bovine C thyrocytes. As the cellular borders of the clustered equine C thyrocytes were faded, it cannot be excluded that at least in some cases, the cytoplasmic processes may have represented sections of cytoplasm from other C thyrocytes. Most importantly, in contrast to the cattle, no clearly visible processes were noted among T thyrocytes in the cross-section. Additional ultrastructural studies are needed to clarify this issue.

Some discrepancies exist regarding the localization of C thyrocytes in horses. While Blähser [22] indicates that equine C thyrocytes appear individually and never form clusters, Yoshikawa et al. [15] report that the C thyrocytes can also aggregate. Similarly, Tanimura et al. [23] note the presence of small groups or large aggregates of C thyrocytes in the equine parafollicular gland. In our present study, equine C thyrocytes were more likely to be found in clusters of various sizes than as separate individuals, and even the C thyrocytes arranged in “chains” were either in direct contact or in very close proximity.

Even less is known about C thyrocytes in cattle. A review by Kameda [2] reports that similarly to porcine cells, bovine C thyrocytes possess a long, slender cytoplasm. However, our present findings indicate that bovine C thyrocytes were polymorphic, ranging in shape from polygonal and oval to irregular; no elongated cells were observed. Similar morphological characteristics of C thyrocytes have been described in European bison and sheep, i.e., species closely related to cattle [12,13]. This is in contrast to the horse specimens, in which many of the polymorphic C thyrocytes were spindle shaped.

A characteristic morphological feature of the bovine C thyrocytes observed in the present study was the presence of obvious, long cytoplasmic processes, which have not been reported in previous studies of this species [34,35]. However, cytoplasmic processes have also been observed in European bison and sheep [12,13].

In our present study, the bovine C thyrocytes were found to be located mainly individually or, less frequently, in small groups consisting of two to five cells. A similar distribution has been noted in sheep [12]; however, in this species, the C thyrocytes tended to form larger groups of four to ten cells, and the clusters were located mainly in the parafollicular position. In contrast, in cattle, the clusters were observed usually within the follicular wall. Similarly, in European bison, C thyrocytes were located either between T thyrocytes or, less commonly, epifollicularly. European bison C thyrocytes adhering to the follicular wall were sometimes clustered in small groups of two to three cells; however, in the areas with higher numbers of C thyrocytes, the clustering was more numerous, forming even solid rims around thyroid follicles [13]. No such C thyrocyte rims were observed in cattle; however, they were noted in horses.

### 4.2. Positioning of C Thyrocytes in Relation to Thyroid Follicle Wall

The specimens were also subjected to combined staining to visualize simultaneously the C thyrocytes and the basement membrane of thyroid follicles. This is the first such co-visualization performed in domestic animals. Previous studies have estimated the localization of C thyrocytes only based on their positioning in relation to T thyrocytes and stromal elements. The localization of the cells was reported according to Sawicki and Kuczyński [5]. Only individual C thyrocytes and their groups that clearly adhered to the outer surface of the basement membrane of thyroid follicles were considered epifollicular. All other C thyrocytes presented within the thyroid stroma were classified as interfollicular. Despite this, it should be noted that at least some of the C thyrocytes appearing as interfollicular in the tangential section may in fact have been epifollicular, or even intrafollicular, as it was not possible to visualize the thyroid follicle basement membrane in this section.

Some discrepancies exist between studies regarding the location of C thyrocytes in relation to thyroid follicles in dogs. While some studies indicate that C thyrocytes are located exclusively outside the follicular wall, in the parafollicular and interfollicular positions [2,7], Rost and Rost [9] report that some cells are located between T thyrocytes within the follicle wall and are in direct contact with the colloid. Our results indicate that the canine C thyrocytes are localized both within and outside the follicular wall; however, surprisingly, the former was more common. Our findings also indicate that the canine C thyrocytes located outside thyroid follicles are positioned epifollicuraly, and that this position was occupied mainly by larger C thyrocyte clusters. However, none of the C thyrocytes were in contact with the colloid.

Fetter and Capen [19] report the existence of both epifollicular and interfollicular C thyrocytes in pigs; however, no such positioning of C thyrocytes was found in the present study. Our observations are in agreement with those of other authors, however scant, who report the presence of C thyrocytes exclusively within the follicular wall in this species [2,11].

Discrepancies exist regarding the localization of C thyrocytes in horses. They have been found to be present within the thyroid follicle wall in some studies [14] and to be located parafollicularly in others [22,23]. Our observations indicate that equine C thyrocytes can occupy both positions; however, the former is more frequent. Moreover, our results indicate that C thyrocytes lying outside the follicular wall are located epifollicularly. These findings are in agreement with those of Yoshikawa et al. [15], who note the presence of C thyrocytes within the follicular wall and interstitial tissue; however, in contrast to our present study, these authors do not provide a detailed location of the C thyrocytes within thyroid stroma.

Little is known about the location of bovine C thyrocytes in relation to thyroid follicles; however, our present findings indicate they can occur in three positions: intrafollicular, epifollicular, and interfollicular. While previous study has also confirmed intrafollicular arrangement [2], the present findings are the first to identify their presence within connective tissue stroma in cattle. A similar distribution pattern of C thyrocytes was recently described in European bison [13], with most being located within the follicular wall, and only single C thyrocytes positioned outside them. Interestingly, older studies conducted on European bison found the C thyrocytes to be located exclusively outside the thyroid follicle wall [36,37]. In sheep, C thyrocytes were also located mainly intrafollicurally, but some were found in perifollicular locations; however, the authors did not provide their exact position within thyroid stroma [12].

In our study, C thyrocytes were observed in three positions: intra-, epi-, and interfollicularly, with the first one being the most common. The intrafollicular localization of C thyrocytes occurred in all analyzed species. The second most frequent localization of C thyrocytes was the epifollicular position presented in all species except pigs, whereas interfollicularly located C thyrocytes were found only in cattle. In contrast, the interfollicular localization of C thyrocytes is frequent in humans. C thyrocytes in this localization are apparently more common than intrafollicular; the second localization of C thyrocytes described in humans [3,38,39,40,41]. Bovine interfollicular C thyrocytes were lying individually, whereas in humans they are usually clustered [40]. In contrast, human C thyrocytes presented within the follicle wall occurred singly or, sometimes as groups with no more than three cells [38]. The presence of C thyrocytes enclosing the follicular epithelium in the form of lamina was also described in humans [40]. This arrangement of C thyrocytes corresponds to the rims of C thyrocytes observed in equine thyroid glands in our study and that described in European bison [13]. However, rims of equine C thyrocytes were located intrafollicularly, whereas in humans they were observed outside the follicular wall [40]. The epifollicular position of human C thyrocytes was not described [3,38,39,40,41]. Interestingly, the results of human studies showed that the localization of C thyrocytes changes with age. In neonates, they are mostly in an intrafollicular position, whereas in adults, interfollicular localization predominates [3,39,41]. Further studies are needed to examine if such phenomenon exist in domestic animals.

### 4.3. Distribution of C Thyrocytes within Thyroid Gland Parenchyma

The location of C thyrocytes within the thyroid lobe parenchyma is not uniform, being located primarily in the center of the thyroid lobe, together with the highest density of active follicles. This is doubtlessly associated with the fact that C thyrocytes are involved in the intrathyroidal regulation of T thyrocyte function by the secretion of various regulatory factors [1]. In contrast, as the thyroid follicles in the subscapular region act mainly as reservoirs of thyroid hormones, there is less need for C thyrocytes in the area. As such, the numbers of C thyrocytes progressively decrease from the central part of the thyroid lobe to its periphery, as well as to its cranial and caudal poles [1,29,30].

To date, differences in C thyrocytes distribution within the thyroid parenchyma have been reported in rodent species, pigs, horses, and European bison [5,6,10,13,15]; however, only three studies have yielded significant findings [10,13,15]. Our present results confirm that the mean number of C thyrocyte profiles increases from the periphery to the central part of the thyroid parenchyma in all analyzed species. This tendency was especially apparent between the superficial and intermediate or central areas of the thyroid lobe, both in individual cases and within particular species; however, this tendency was not so evident when intermediate and central areas were analyzed in individual cases. In at least some animals belonging to each species, the differences in fractions of C thyrocytes between these regions of the thyroid lobe did not exceed 1–2%. In addition, some horses and cattle demonstrated a higher percentage of C thyrocyte profiles in the intermediate area than in the central area. This lack of consistency between individuals can be explained by the fact that C thyrocyte distribution is not homogeneous within the thyroid gland, as the cells tend to form clumps of various sizes surrounded by areas free of C thyrocytes; this heterogeneity would result in differences in C thyrocyte profile numbers between visual fields. However, when all cases from particular species were analyzed together, significant differences in fraction of C thyrocytes were found between the three examined areas of the thyroid lobe.

### 4.4. Number of C Thyrocytes within Thyroid Gland Parenchyma

Regarding the percentage of C thyrocyte profiles in relation to the profiles of total endocrine cell population, dogs, pigs, and cattle demonstrated similar total fractions of C thyrocytes (approximately 8–9%), and these were only slightly higher than in horses (approximately 6%). These results are in line with previous reports in other species, i.e., ranging from less than 1% in humans [1] to approximately 10–20% in laboratory animals [4,5,42]. Only in dogs the percentage of C thyrocytes was apparently higher, estimated between 20–45% [42].

Until now, dogs were considered to have the highest percentage of C thyrocytes among all animal species [42]. However, our results indicated a similar fraction of C thyrocytes to other species, only exceeding 17% in one of 20 cases. These discrepancies result probably from the fact that Kameda [42] focused on areas of thyroid glands with remnants of ultimobranchial bodies, characterized by considerably larger numbers of C thyrocytes, as mentioned previously.

In pigs, the percentage of C thyrocytes has been found to vary from approximately 1% [11] to 4% [10]. Our results indicate this value to be approximately 8%. Whereas the present study only determined the percentage of C thyrocyte profiles in the midsection of the thyroid lobe, Tsuchiya et al. [10] analyzed nine sections cut throughout the whole gland. Taking into account that C thyrocytes are located mainly in the medial part of the thyroid lobes, the differences between our results and the study by Tsuchiya et al. [10] are easy to explain. Unfortunately, Young et al. [11] give no information regarding the methods of C thyrocyte calculation.

Our findings are the first to provide fractions of C thyrocytes in horses and cattle. Only one previous study has analyzed the percentage of C thyrocytes in European bison [13]. While European bison and cattle have a similar percentage of C thyrocytes, slightly higher levels are seen in cattle (8.58% vs. 7.33%). This could indicate the presence of inter-species differences, especially considering that both the present study and the previous one used the same method of C thyrocyte counting, i.e., the same area of the thyroid lobe, and the same counting procedure.

Our results, although based on the number of C thyrocyte profiles calculated only in the medial part of the thyroid lobe, indicate that the differences in the percentage of C thyrocytes within the thyroid gland are not pronounced and that most extensive inter-species discrepancies are probably associated with the method of calculation used in studies and the choice of thyroid gland area. Both our present work and that on European bison [13] were performed in the medial part of the thyroid lobe, the region containing the most numerous C thyrocytes, and showed a high consistency of the results between the analyzed species.

Some studies indicate that the shapes, sizes, and distribution patterns of C thyrocytes are species-specific [2,15,22]. In the light of our results and their confrontation with limited data presented in the available literature, these observations seem more understandable. Some variations in morphology and distribution pattern of C thyrocytes indeed exist between species. However, these variations tend to be observed only between taxonomically distant species and only slight differences are observed between closely related taxa, such as ruminants.

The study serves detailed microscopic characteristics of C thyrocytes in healthy animals. It can be valuable for further studies of C thyrocytes, especially in terms of their paracrine functions. Moreover, the description of normal C thyrocytes morphology and distribution pattern can also serve as a reference for further microscopic studies on C thyrocytes in pathological conditions, such as C thyrocyte hyperplasia or medullary carcinoma.

## 5. Conclusions

This study provides the first detailed comparison of C thyrocyte morphology, distribution within the thyroid gland parenchyma, and position in relation to thyroid follicles between four domestic animals. Some previously suggested features have been confirmed, complemented, and clarified, and some unique features have been reported for the first time. Our results indicate that the studied species of *Canidae*, *Suidae*, *Bovidae,* and *Equidae*, carried their unique morphological characteristic and distribution pattern of C thyrocytes that can be easily recognized microscopically; however, the total numbers of C thyrocyte profiles and their localization within the thyroid lobe are comparable. The study not only broadens the current knowledge of C thyrocytes but can also be valuable for future studies on this subject in both physiological and pathological conditions.

## Figures and Tables

**Figure 1 animals-12-01324-f001:**
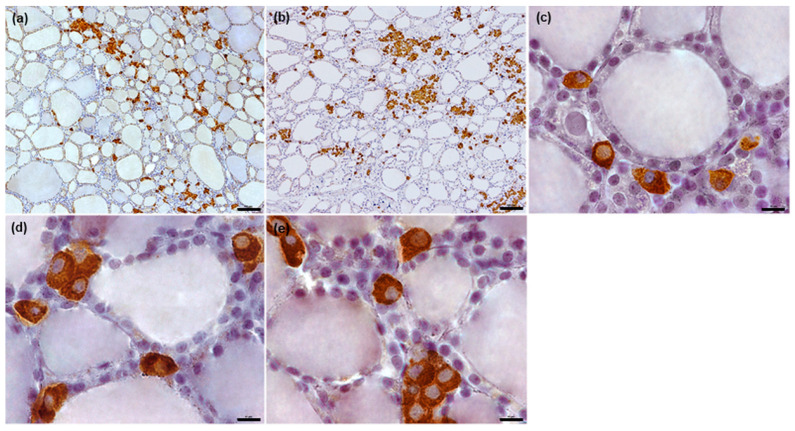
Representative images of canine C thyrocytes arrangement and morphology. (**a**,**b**): Section of thyroid parenchyma with C thyrocytes arranged in groups of various sizes. (**c**–**e**): Typical morphology of C thyrocytes in dogs. Immunoperoxidase staining with anti-calcitonin antibody, scale bars: (**a**,**b**) = 100 μm, (**c**–**e**) = 10 μm.

**Figure 2 animals-12-01324-f002:**
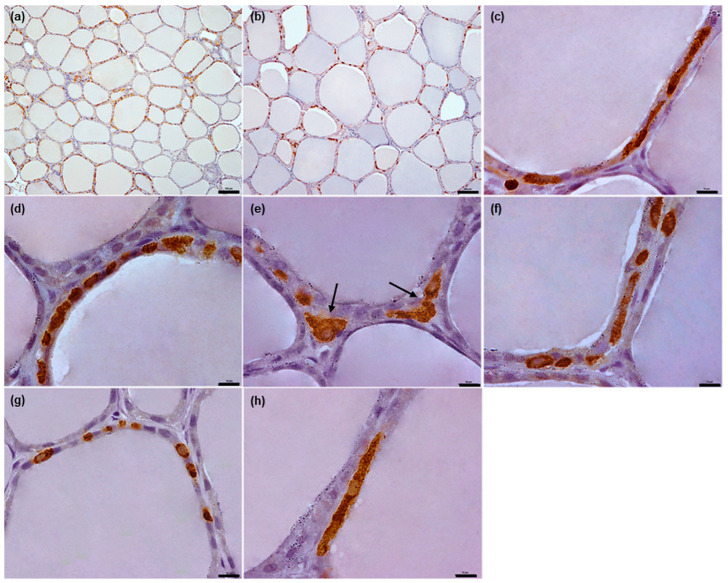
Representative images of porcine C thyrocytes arrangement and morphology. (**a**,**b**): Typical distribution of C thyrocytes in thyroid parenchyma. (**c**–**f**): C thyrocyte arrangement within thyroid follicles: “chains” (**c**,**d**), small groups ((**e**), arrows), and cells located separately in various orientations in relation to the surface of the thyroid follicle (**f**). (**g**,**h**): Typical morphology of C thyrocytes in pigs: cells visible in transverse (**g**) and longitudinal (**h**) sections. Immunoperoxidase staining with anti-calcitonin antibody, scale bars: (**a**,**b**) = 100 μm, (**c**–**h**) = 10 μm.

**Figure 3 animals-12-01324-f003:**
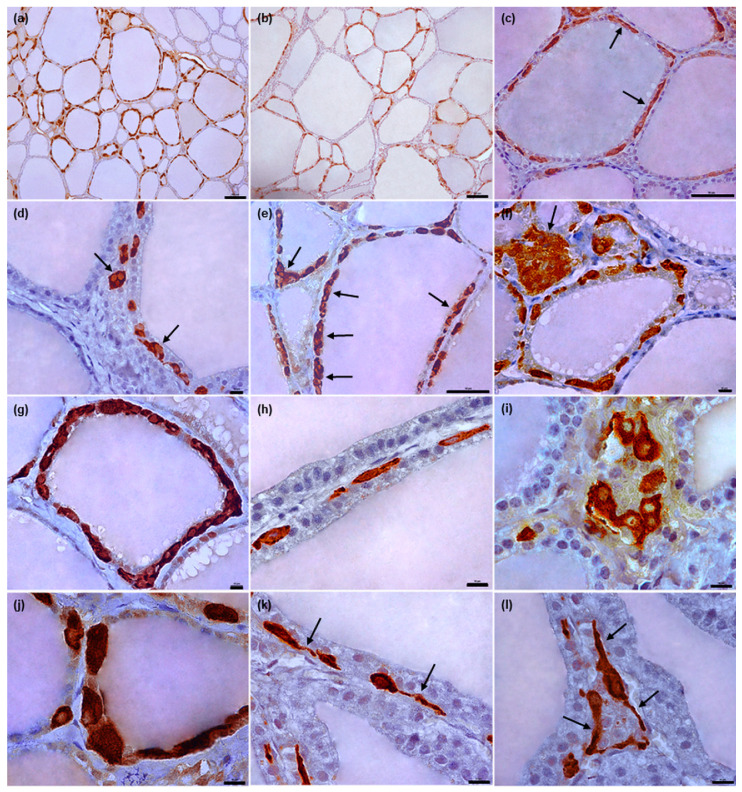
Representative images of equine C thyrocytes arrangement and morphology. (**a**,**b**): Typical distribution of C thyrocytes in thyroid parenchyma. (**c**–**g**): Typical arrangement of C thyrocytes in thyroid follicles: cells located separately or in “chains” ((**c**), arrows), in groups of various sizes ((**d**–**f**), arrows), and as a rim encompassing the whole follicle (**g**). (**h**–**j**): Typical morphology of C thyrocytes in horses. (**k**,**l**): C cells with slender endings of cytoplasm (arrows). Immunoperoxidase staining with anti-calcitonin antibody, scale bars: (**a**,**b**) = 100 μm, (**c**,**e**) = 50 μm, (**d**,**f**–**l**) = 10 μm.

**Figure 4 animals-12-01324-f004:**
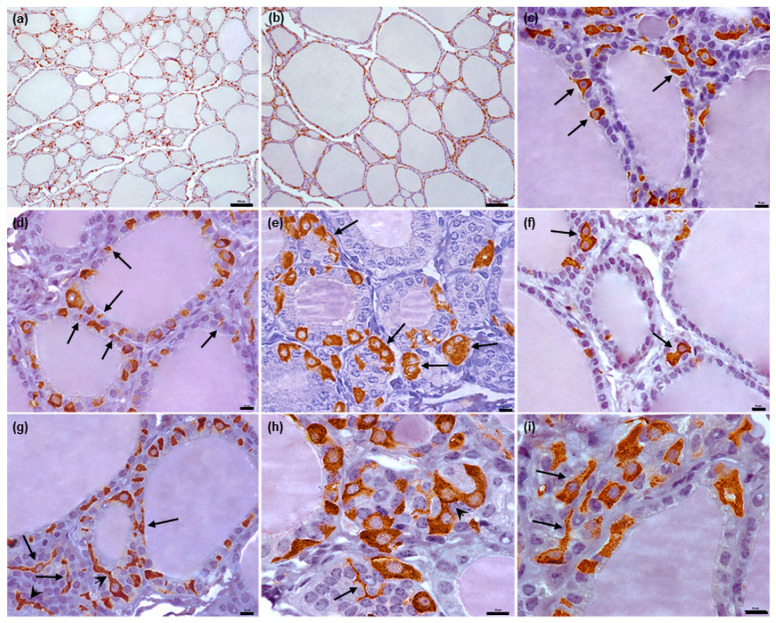
Representative images of bovine C thyrocytes arrangement and morphology. (**a**,**b**): Typical distribution of C thyrocytes in thyroid parenchyma. (**c**): Single C thyrocytes reach the apical surface of T thyrocytes (arrows). (**d**–**f**): Typical arrangement of C thyrocytes in thyroid follicles: cells located separately, cross-sections thorough cytoplasmic processes are visible ((**d**), arrows) or in small groups ((**e**–**f**), arrows). (**g**–**i**): Typical morphology of C thyrocytes in cattle, some cells possess long cytoplasmic processes (arrows) and contact with other C thyrocytes via these protrusions (arrowheads). Immunoperoxidase staining with anti-calcitonin antibody, scale bars: (**a**,**b**) = 100 μm, (**c**–**i**) = 10 μm.

**Figure 5 animals-12-01324-f005:**
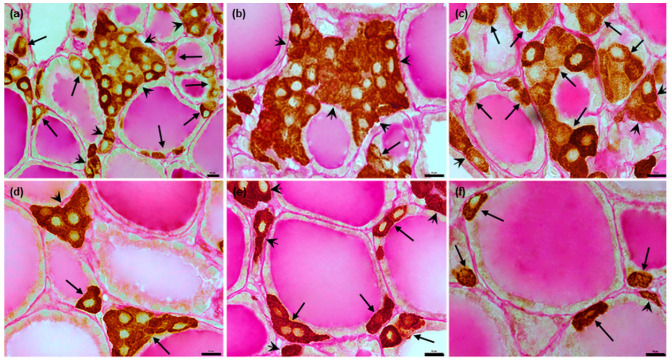
Representative images of canine C thyrocytes position in relation to the follicular wall. (**a**,**b**): Very large groups of C thyrocytes located epifollicularly. (**c**–**f**): Smaller cell groups and single cells in both intrafollicular and epifollicular positions. Single C thyrocytes or their groups located intrafollicularly are marked with arrows and those located epifolliculary with arrowheads. Combined anti-calcitonin antibody immunohistochemistry–periodic acid Schiff method, scale bars: (**a**–**f**) = 10 μm.

**Figure 6 animals-12-01324-f006:**
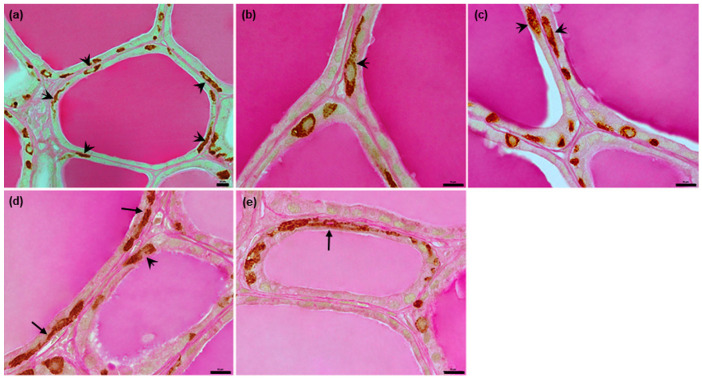
Representative images of porcine C thyrocytes position in relation to the follicular wall. (**a**–**e**): All C thyrocytes lie exclusively in the intrafollicular position. They are scattered irregularly around the follicle perimeter, separately or in “chains” (arrows), and are visible in both longitudinal (arrowheads) and transverse sections. Combined anti-calcitonin antibody immunohistochemistry–periodic acid Schiff method, scale bars: (**a**–**e**) = 10 μm.

**Figure 7 animals-12-01324-f007:**
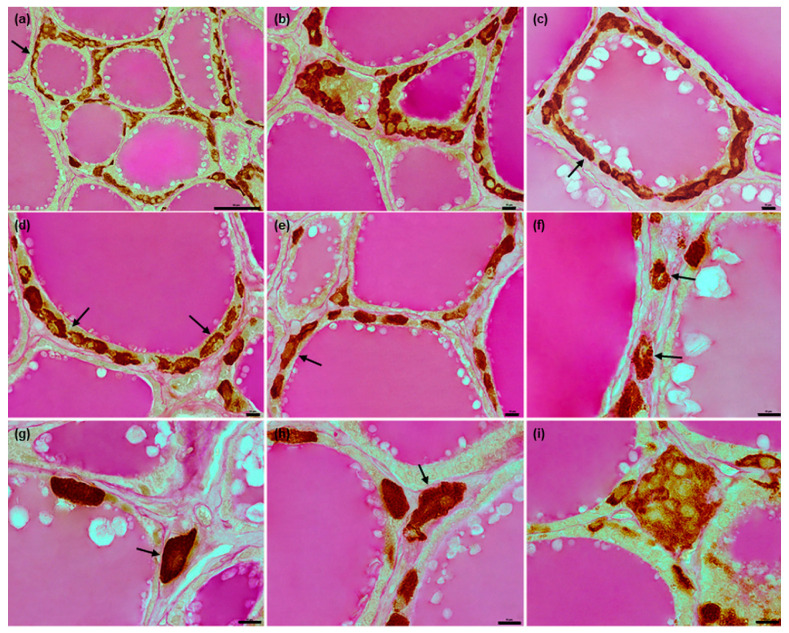
Representative images of equine C thyrocytes position in relation to the follicular wall. (**a**–**e**): C thyrocytes located only in the intrafollicular position: in the form of large groups or rims (arrows) around the perimeter of follicles (**a**–**c**), as “chains” (arrows) or single cells scattered irregularly between T thyrocytes (**d**–**e**). (**f**–**i**): C thyrocytes in epifollicular position, located individually ((**f**–**h**), arrows) or in a large group (**i**). Combined anti-calcitonin antibody immunohistochemistry–periodic acid Schiff method, scale bars: (**a**) = 50 μm, (**b**–**i**) = 10 μm.

**Figure 8 animals-12-01324-f008:**
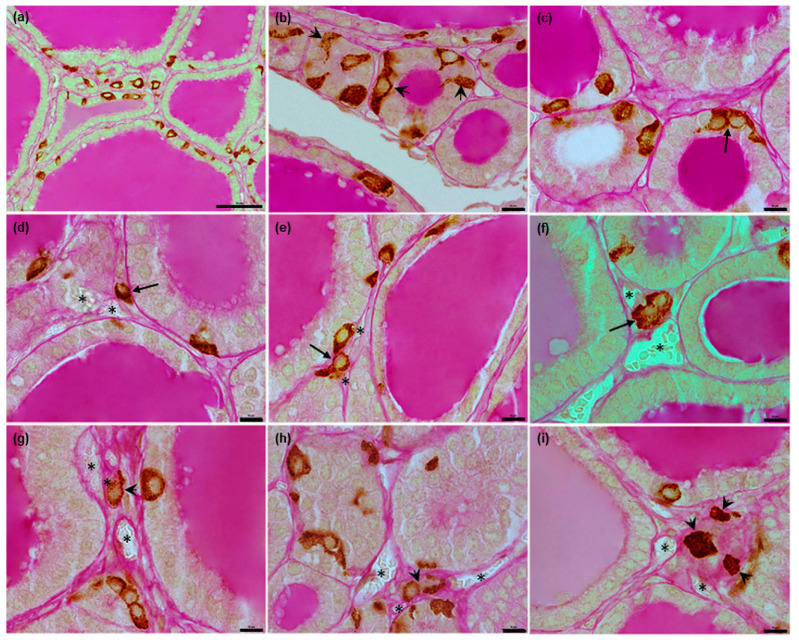
Representative images of bovine C thyrocytes position in relation to the follicular wall. (**a**–**c**): C thyrocytes located in intrafollicular position: in the form of single cells scattered irregularly between T thyrocytes (**a**,**b**) or in small groups ((**c**), arrow), some cells possess cytoplasmic processes (arrowheads). (**d**–**i**): C thyrocytes located outside follicular wall: epifollicularly (arrows), separately (**d**–**e**) or in small group (**f**), and interfollicularly (arrowheads), separately (**g**) or in small groups (**h**,**i**), C cells located outside thyroid follicles frequently lie adjacent to blood capillaries (asterisks). Combined anti-calcitonin antibody immunohistochemistry–periodic acid Schiff method, scale bars: (**a**) = 50 μm, (**b**–**i**) = 10 μm.

**Table 1 animals-12-01324-t001:** Fraction of C thyrocytes in three different regions of the medial part of the thyroid lobe and total fraction of C thyrocytes in examined species.

Species	Fraction of C Thyrocytes (%)			Total Fraction of C Thyrocytes (%)
	Superficial Area	Central Area	Intermediate Area	
	Mean ± SD (Range of Values)	Mean ± SD (Range of Values)	Mean ± SD (Range of Values)	Mean ± SD
Dogs	2.55 ± 5.32 (0–35.33)	12.48 ± 9.42 (0–44.75)	8.43 ± 9.29 ^a^ (0–47.18)	8.87 ± 3.72
Pigs	3.81 ± 5.12 (0–26.09)	10.79 ± 6.15 (0–29.09)	7.66 ± 5.49 ^a^ (0–24.17)	7.98 ± 2.94
Horses	1.55 ± 3.73 (0–27.69)	8.87 ± 7.78 (0–33.67)	7.44 ± 8 ^a^ (0–42.46)	5.83 ± 2.56
Cattle	2.62 ± 4.7 (0–26.67)	12.96 ± 7.97 (0–32.57)	10.75 ± 7.43 ^a^ (0–34.74)	8.58 ± 2.08

^a^—The differences between particular regions of the thyroid lobe in each species are significant (*p* ≤ 0.001).

## Data Availability

The datasets generated during and/or analyzed during the current study are available from the corresponding author on reasonable request.

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
