# Peer review of "Comparative Histology of C Thyrocytes in Four Domestic Animal Species: Dog, Pig, Horse, and Cattle"

_animals, 2022, doi:10.3390/ani12101324_

Round 1

Reviewer 1 Report

The rationale of the study is a strong one as the distribution of C-thyrocytes in various species has not been mapped yet. The manuscript would deserve to be published after addressing several issues to make the paper more beneficial to the Journal readers.

-Include the types of staining and antigens and the quantitative parameters into the abstract. "Morphometric analysis" on line 26 is not specific enough.

-Abstract, line 31 - no differences among the species?

-Abstract, line 32: species in plural, so should be the pronoun, its-->theirs, right?

-Line 39: ...is composed... -->contains; it is definitely composed of more cell types

-Terminologia histologica differenciates two types of thyrocytes: T thyrocyte and C thyrocyte (C cell, parafollicular cell). Use this terminology consistently in the whole paper. Mentioning just thyrocytes (line 39, 43 etc.) is not specific enough

-Line 97: How were the sections selected? How meny sections were evaluated per animal?

-line 115, 118: shiff method --> Schiff method; also, be speficic about your own modification mentioned on line 115

-I have objections against the terminology on counting the "C cell concentration". First, you did not count the cells. Cells are three-dimensional objects and their dimensions are not negligible when related to the section thickness and depth of focus. What you counted were the C thyrocytes cell profiles and T thyrocytes cell profiles. Second, the ratio you have calculated on line 128 i a fraction of C thyrocytes profiles within the number of all thyrocytes profiles. This should reflect the terminology used by stereology. For details, see any textbook on stereology (e.g. ISBN: 0–8018–6797–5) and papers setting standards for quantitative morphometric studies (e.g. https://pubmed.ncbi.nlm.nih.gov/23769130/).

-How many cells did you count per one case, i.e. one area of a single animal? You measured 40 C cells, but what number of cells was the counting of the C cells fraction based on?

-line 151 Differences-->differences

The scale bars in all figures are barely visible, consider increasing the thickness of the bars, please. The Figures 5-6 migh benefit from correction of the white balance. Rearranging Figure 6 into a 2x2 composite image would use the space in a more efficient way.

-The Figures 5-6 that are supposed to demonstrate the position of C cells relatively to the basal membrane, would really benefit from using a more detailed magnification as the mutual position of cells and basal membrane is barely visible to me.

-Section 3.1. The biological meaning of the morphometric parameters, such as area, perimeter, length, width, and circularity is obscure, especially in formalin fixed paraffin embedded sections. As far as I am aware, these formal morphometric parameters have no meaning or interpretation in C thyrocytes, or? How should the reader understand this part of your results? How can you interpret the measurements done of two-dimensional cell profiles of three-dimensional objects (cells)? Do you realize that the shapes of cell profiles strongly depend on the section plane and anisotropic orientation of the cells on your sections? The cells are definitely far from having a spherical shape, which is the only shape that would not be biased in your study design.

-The Discussion is really lengthy. Please divide it into multiple subsections, each with a clearly understandable message. Moreover, compare your findings with the information previously published on the position of C thyrocytes in relation to the basal membrane in humans.

-Try to formulate clear and succing study implications at the end of the Discussion.

Author Response

We would like to thank the Reviewer for appreciating our work and the helpful comments. We did our best to address all comments and we hope that the manuscript has improved.

-Include the types of staining and antigens and the quantitative parameters into the abstract. "Morphometric analysis" on line 26 is not specific enough.

The staining and antigen has been added:

C thyrocytes were visualized immunohistochemically with anti-calcitonin rabbit polyclonal antibody alone and combined with periodic acid Schiff method to visualize simultaneously C thyrocytes with the basal membranes of thyroid follicles.

We have also added numerical data regarding fractions of C thyrocytes:

The mean total fraction of C thyrocytes in the superficial, intermediate and central areas were as follows: 2.55%, 8.43% and 12.48% in dogs; 3.81%, 7.66% and 10.79% in pigs; 1.55%, 7.44% and 8.87% in horses; 2.62%, 10.75% and 12.96% in cattle. No statistical differences in total number of C thyrocyte profiles were observed among species (8.87% in dogs, 8.58% in cattle, 7.98% in pigs and 5.83% in horses).

After Reviewer remark regarding “Morphometric analysis” we have decided to remove this part of our study (see below). Thus, the sentence regarding morphometric analysis has been removed from the Abstract.

-Abstract, line 31 - no differences among the species?

All these values were not statistically significant. To be more clear we have added word “statistical”:

No statistical differences in total number of C thyrocyte profiles were observed among species.

-Abstract, line 32: species in plural, so should be the pronoun, its-->theirs, right?

We are sorry for this mistake. It has been corrected.

-Line 39: ...is composed... -->contains; it is definitely composed of more cell types

It has been corrected accordingly:

The thyroid gland contains two types of endocrine cells

-Terminologia histologica differenciates two types of thyrocytes: T thyrocyte and C thyrocyte (C cell, parafollicular cell). Use this terminology consistently in the whole paper. Mentioning just thyrocytes (line 39, 43 etc.) is not specific enough

Nomina Histologica Veterynaria (2017) recognize thyroid endocrine cells as: follicular thyrocytes and parafollicular thyrocytes, however terms: parafollicular cells and C cells are also acceptable. We would favour to define thyroid endocrine cells as thyrocytes and C cells as these terms, especially C cells, are widely used in literature: scientific and popular articles, as well as textboxes. We have changed the terminology throughout the whole article according to the Reviewer’s suggestion and to avoid confusion for readers who are less adept at histological nomenclature we have reworded the first sentence of Introduction section as below, and used these terms consistently in whole paper:

The thyroid gland contains two types of endocrine cells: T thyrocytes (follicular thyrocytes, thyrocytes) and C thyrocytes (C cells, parafollicular cells); however, C thyrocytes constitute only a minor proportion of the total thyroid parenchyma

-Line 97: How were the sections selected? How meny sections were evaluated per animal?

In each case, we collected whole thyroid lobe. Then, the lobe was cut transversely into two halves. Then, the sample of proximal part was used for the further analysis. After embedding it in paraffin two tissue specimens separated from each other by a distance of 100 µm were cut along transverse axis of the lobe surface in midsection and evaluated. Quantitative analysis of C thyrocytes included counting of their profiles in 14 visual fields per each area (superficial, intermediate and central) per each tissue section.

We have not added these details due to remain the manuscript shorter, however, we can add them if the Reviewer finds it important.

-line 115, 118: shiff method --> Schiff method; also, be speficic about your own modification mentioned on line 115

It was a typing error and it has been corrected.

Our modification was a combination of two separated staining methods: immunohistochemical staining with periodic acid Schiff reaction in one staining procedure. It has been added to the text:

After immunohistochemical detection with anti-calcitonin antibodies, these specimens were subjected to further staining by the periodic acid Schiff method (own modification related to the use of these two methods in one staining procedure). Briefly, the reaction with the primary antibody was visualized using the DAB solution, following which, the slides were rinsed with tap water for 10 minutes and placed in distilled water for 5 minutes. They were then subjected to staining by the routine periodic acid Schiff method

-I have objections against the terminology on counting the "C cell concentration". First, you did not count the cells. Cells are three-dimensional objects and their dimensions are not negligible when related to the section thickness and depth of focus. What you counted were the C thyrocytes cell profiles and T thyrocytes cell profiles. Second, the ratio you have calculated on line 128 i a fraction of C thyrocytes profiles within the number of all thyrocytes profiles. This should reflect the terminology used by stereology. For details, see any textbook on stereology (e.g. ISBN: 0–8018–6797–5) and papers setting standards for quantitative morphometric studies (e.g. https://pubmed.ncbi.nlm.nih.gov/23769130/).

In this part of our investigation we adapted method of C thyrocytes assessment from paper of Tsuchiya et al. (Immunohistochemical study on the C cells in pig thyroid glands, Acta Anat. (Basel), 1984, 120:138-141) and we followed the terminology used be these authors. Moreover, authors of other papers assessing quantitatively the number of C thyrocytes within thyroid based on their two-dimensional cell profiles used term “cells” instead of “cell profiles” (i.e. .Yoshikawa et al.: Distribution of C cells in thyroids and association with age and sex in racing horses. J Equine Sci. 2001, 12:39–45). We absolutely agree that from stereological point of view this terminology is not correct, thus we have changed it and we hope it is more accurate now:

In each visual field, the overall number of C and T thyrocytes profiles was calculated. Following this, the results from each area were summarized, and the proportion of C thyreocytes profiles to profiles of total endocrine cell population (fraction of C thyrocytes) was calculated according to the following formula:

Fraction of C thyroids [%] = number of C thyrocytes profiles/number of all endocrine cells profiles (C and T thyrocytes) x 100%

In each case, the fraction of C thyrocytes was scored in both tissue sections originating from each lobe and the mean value was calculated. The numbers were recorded for each area of the thyroid lobe and for the whole tissue section (relative total fraction of C thyrocytes).

The terms: “fraction of C thyrocytes” and “number of C thyrocytes profiles” instead of “C cell concentration” and “C cell number” has been used in whole text.

-How many cells did you count per one case, i.e. one area of a single animal? You measured 40 C cells, but what number of cells was the counting of the C cells fraction based on?

We counted all profiles of T and C thyrocytes presented within each visual field. Usually each visual field corresponded to approximately 100 endocrine cells. Then, the percentage of C thyreocytes profiles was calculated for each area (i.e. superficial, intermediate and central area) per animal.

We have not added all these details due to remain the manuscript shorter, however, we can add them if the Reviewer finds it important

-line 151 Differences-->differences

This typing error has been corrected.

The scale bars in all figures are barely visible, consider increasing the thickness of the bars, please. The Figures 5-6 migh benefit from correction of the white balance. Rearranging Figure 6 into a 2x2 composite image would use the space in a more efficient way.

The scale bars has been corrected. Regarding figures, please see our comment below.

-The Figures 5-6 that are supposed to demonstrate the position of C cells relatively to the basal membrane, would really benefit from using a more detailed magnification as the mutual position of cells and basal membrane is barely visible to me.

We have increased  the figure size, and we hope that Editor will accept it. Moreover we replaced most of figures stained with Schiff method taken at x400 or x600 magnification with figures taken at magnification x1000. We hope that it improved the chance for better analysis.

The following changes has been made in fig. 5-8:

  • 5: the photographs a,c,d,e have been replaced by figures taken at magnification x1000; the photograph b has been moved into a position.
  • 6: the photographs a,b,d have been replaced by figures taken at magnification x1000; the photograph c has been moved into a position; additional figure e taken at magnification x1000 has been added
  • 7: the photographs a,c,d,e,f,g,h,i have been replaced by figures taken at magnification x600 (a,c,d,e) or x1000 (f,g,h,i); the photograph b has been moved into a position
  • 8: the photographs a,c,d,e,f,g,h,i have been replaced by figures taken at magnification x1000; the photograph b has been moved into a position

-Section 3.1. The biological meaning of the morphometric parameters, such as area, perimeter, length, width, and circularity is obscure, especially in formalin fixed paraffin embedded sections. As far as I am aware, these formal morphometric parameters have no meaning or interpretation in C thyrocytes, or? How should the reader understand this part of your results? How can you interpret the measurements done of two-dimensional cell profiles of three-dimensional objects (cells)? Do you realize that the shapes of cell profiles strongly depend on the section plane and anisotropic orientation of the cells on your sections? The cells are definitely far from having a spherical shape, which is the only shape that would not be biased in your study design.

We absolutely agree that the shapes of cell profiles strongly depend on the section plane and anisotropic orientation of the cells on examined sections. We undertake this analysis to characterize the C thyrocytes in as much detail as possible and to enrich our morphological observations with additional numerical data. Taking into account that this morphometric analysis does not contribute relevant information into the current study and can serve as a source of confusion we propose to remove this part (i.e. sections 2.3 and 3.4 (by the copy/paste mistake all subsection of the Result part have the same number – we are sorry for this), together with fig.9). Especially, that due to the length of the text, we resigned from any comment this part of the research in the Discussion section. We hope that, this way we will avoid any biases and improve the quality of the study.

-The Discussion is really lengthy. Please divide it into multiple subsections, each with a clearly understandable message. Moreover, compare your findings with the information previously published on the position of C thyrocytes in relation to the basal membrane in humans.

We have divided the text into the following subsections:

4.1 Morphological characteristic and distribution pattern of C thyrocytes

4.2 Positioning of C thyrocytes in relation to thyroid follicle wall

4.3 Distribution of C thyrocytes within thyroid gland parenchyma

4.4 Number of C thyrocytes within thyroid gland parenchyma

Regarding comparing our findings with the information previously published on the position of C thyrocytes in relation to the basal membrane in humans, we have added the following text at the end of subsection 4.2 Positioning of C thyrocytes in relation to thyroid follicle wall:

In our study C thyrocytes were observed in three positions: intra-, epi- and interfollicularly with the first one being the most common. The intrafollicular localization of C thyrocytes was occurred in all analysed species. The second most frequent localisation of C thyrocytes was epifollicular position presented in all species except pigs, whereas interfollicularly located C thyrocytes were found only in cattle. In contrast, interfollicular localization of C thyrocytes is frequent in humans. C thyrocytes in this localization are apparently more common than intrafollicular; the second localization of C thyrocytes described in humans (Hazard, 1977; Das et al., 2017; Lin, 1983; Wolfe et al, 1974 and 1975). Bovine intrafollicular C thyrocytes were lying individually, whereas in human they usually clustered (Lin, 1983). In contrast, human C thyrocytes presented within follicle wall occurred singly or, sometimes as a groups not more than three cells (Hart, 1977). Presence of C thyrocytes enclosing the follicular epithelium in the form of lamina was also described in humans (Lin, 1983). This arrangement of C thyrocytes corresponds to the rims of C thyrocytes observed in equine thyroid gland in our study and that described in European bison. However rims of equine C thyrocytes were located intrafollicularly, whereas in humans they were observed outside the follicular wall (Lin, 1983). Epifollicular position of human C thyrocytes was not described (Hazard, 1977; Das et al., 2017; Lin, 1983; Wolfe et al, 1974 and 1975). Interestingly, results of human studies showed, that localization of C thyrocytes changes with age. In neonates they are mostly in intrafollicular position, whereas in adults interfollicular localization predominates (Das et al., 2017; Wolfe et al, 1974 and 1975). Further studies are needed to examine if such phenomenon exist in domestic animals.

References:

  • Hazard, J.B. The C cells (parafollicular cells) of the thyroid gland and medullary thyroid carcinoma. A review. J. Pathol. 1977, 88, 213–250.
  • Das, S.S., Mishra, S., Kaul, J.M. Development of parafollicular Cells and their relationship with developing thyroid follicles in human foetuses. J Clin Diagn Res. 2017, 11, AC01–AC04. doi: 10.7860/JCDR/2017/26211.10225.
  • Lin, Y.T Immunoperoxidase staining studies on C-cells in fresh human thyroid. Nihon Naibunpi Gakkai Zasshi. 1983, 59, 1244-1255. doi: 10.1507/endocrine1927.59.9_1244.
  • Wolfe, H.J., Voelkel, E.F., Tashjian, A.H. Distribution of calcitonin-containing cells in the normal adult human thyroid gland: a correlation of morphology with peptide content. J Clin Endocrinol Metab. 1974, 38, 688–694. doi: 10.1210//jcem-38-4-688.
  • Wolfe, H.J, DeLellis, R.A., Voelkel, E.F., Tashjian, A.H. Distribution of calcitonin-containing cells in the normal neonatal human thyroid gland: a correlation of morphology with peptide content. J Clin Endocrinol Metab. 1975, 41, 1076-1081. doi: 10.1210/jcem-41-6-1076.

-Try to formulate clear and succing study implications at the end of the Discussion.

We have added the following information at the end of the Discussion section:

The study serves detailed microscopic characteristic of C thyrocytes in healthy animals. It can be valuable for further studies of C thyrocytes, especially in terms of their paracrine functions. Moreover, description of normal C thyrocytes morphology and distribution pattern can also serve as a reference for further microscopic studies on C thyrocytes in pathological conditions, such as C thyrocyte hyperplasia or medullary carcinoma.

Reviewer 2 Report

The reviewed article is a valuable contribution to our knowledge about morphology and distribution of C cells in the thyroid gland. Article is well organized and is a example of professional morphological work. In my opinion only the minor revision carried out by English native speaker experienced with scientific works is needed.

Author Response

We would like to thank the Reviewer for appreciating our work and such encouraging comment. we will do our best to improve also English and check all text again.

Reviewer 3 Report

The article is interesting. Although the presence of C cells has already been described, this article carries out a systematic study in a large number of specimens with a complete statistical study.

The main comment relates to the use of the PAS staining. The purpose of this study is to determine if C cells share the basal lamina with thyrocytes, that is, to determine if they are embedded in the epithelium or if they are found in the connective stroma. The photographs shown with the PAS technique are abundant, but at low magnification, so the basal lamina is not appreciated. It is necessary to display images at the maximum magnification possible. The current ones are of such low magnification that they are equivalent to the figures in which only immunohistochemical staining has been performed and therefore do not provide any additional information. Also in some images it is pointed out that the C cells have cytoplasmic processes, but the magnification of the image is low and it is not seen; images at maximum magnification are also needed to see these extensions.

In the M&M it is necessary to indicate the fixation time of the samples.

Author Response

We would like to thank the Reviewer for appreciating our work and the helpful comments. We did our best to address all comments and we hope that the manuscript has improved.

The main comment relates to the use of the PAS staining. The purpose of this study is to determine if C cells share the basal lamina with thyrocytes, that is, to determine if they are embedded in the epithelium or if they are found in the connective stroma. The photographs shown with the PAS technique are abundant, but at low magnification, so the basal lamina is not appreciated. It is necessary to display images at the maximum magnification possible. The current ones are of such low magnification that they are equivalent to the figures in which only immunohistochemical staining has been performed and therefore do not provide any additional information. Also in some images it is pointed out that the C cells have cytoplasmic processes, but the magnification of the image is low and it is not seen; images at maximum magnification are also needed to see these extensions.

We have increased will ask to increase the figure size, and we hope that Editor will accept it. Moreover we replaced most of figures stained with Schiff method taken at x400 or x600 magnification with figures taken at magnification x1000. We hope that it improved the chance for better analysis.

The following changes has been made in fig. 5-8:

  • 5: the photographs a,c,d,e have been replaced by figures taken at magnification x1000; the photograph b has been moved into a position.
  • 6: the photographs a,b,d have been replaced by figures taken at magnification x1000; the photograph c has been moved into a position; additional figure e taken at magnification x1000 has been added
  • 7: the photographs a,c,d,e,f,g,h,i have been replaced by figures taken at magnification x600 (a,c,d,e) or x1000 (f,g,h,i); the photograph b has been moved into a position
  • 8: the photographs a,c,d,e,f,g,h,i have been replaced by figures taken at magnification x1000; the photograph b has been moved into a position

Unfortunately, photographs from figures 3 and 4 that represents cells with processes (Fig.3k-l and Fig.4h-i) were taken at magnification x1000, thus we are not able to do more than increase the whole image size to improve their readability. We hope that Editor will accept it.

In the M&M it is necessary to indicate the fixation time of the samples.

The fixation time was 24 hours. This information has been added to the relevant sentence:

Tissue specimens were fixed in 10% neutral buffered formalin for 24 hours, processed by the common paraffin technique, and cut into 3 µm sections.

Round 2

Reviewer 3 Report

It is ok